# A new method of Bayesian causal inference in non-stationary environments

**Shuji Shinohara**[1]*, **Nobuhito Manome**[1,2], **Kouta Suzuki**[1,2], **Ung-il Chung**[1],
**Tatsuji Takahashi**[1,3], **Hiroshi Okamoto**[1], **Yukio Pegio Gunji**[4], **Yoshihiro Nakajima**[5],
**Shunji Mitsuyoshi**[1]

**1** Department of Bioengineering, Graduate School of Engineering, The University of Tokyo, Tokyo, Japan,
**2** Department of Research and Development, SoftBank Robotics Group Corp., Tokyo, Japan, **3** School of
Science and Engineering, Tokyo Denki University, Saitama, Japan, **4** Department of Intermedia Art and
Science, School of Fundamental Science and Technology, Waseda University, Tokyo, Japan, **5** Graduate
School of Economics, Osaka City University, Osaka, Japan

* shinohara@bioeng.t.u-tokyo.ac.jp

pone.0233559

UNITED KINGDOM

**Data Availability Statement:** All data files are
available from the Dryad database (https://
datadryad.org/stash/share/

## Abstract

Bayesian inference is the process of narrowing down the hypotheses (causes) to the one
that best explains the observational data (effects). To accurately estimate a cause, a consid-
erable amount of data is required to be observed for as long as possible. However, the
object of inference is not always constant. In this case, a method such as exponential mov-
ing average (EMA) with a discounting rate is used to improve the ability to respond to a sud-
den change; it is also necessary to increase the discounting rate. That is, a trade-off is
established in which the followability is improved by increasing the discounting rate, but the
accuracy is reduced. Here, we propose an extended Bayesian inference (EBI), wherein
human-like causal inference is incorporated. We show that both the learning and forgetting
effects are introduced into Bayesian inference by incorporating the causal inference. We
evaluate the estimation performance of the EBI through the learning task of a dynamically
changing Gaussian mixture model. In the evaluation, the EBI performance is compared with
those of the EMA and a sequential discounting expectation-maximization algorithm. The
EBI was shown to modify the trade-off observed in the EMA.

## Introduction

The aim of Bayesian inference is to deduce the hidden cause behind observed data by retro-
spectively applying statistical inferences. The relationship between Bayesian inference and
brain function has attracted significant attention in recent years in the field of neuroscience
[1,2]. In Bayesian inference, the degree of confidence for each hypothesis is updated based on
a predefined model for each hypothesis by incorporating the current observational data. In
other words, Bayesian inference is a process of narrowing down hypotheses (causal candidates)
to one that best explains the observational data (the effects).

As an example, consider a situation in which one attempts to read another's emotions.
Because one cannot directly view another's emotions, they can only be inferred from external

X4r1OvPYvzE7Uc1drn5p-W6KMqDNzqrx8OJrHbITUvQ).

**Funding:** This research is supported by the Center of Innovation Program from the Japan Science and Technology Agency, JST and by JSPS KAKENHI Grant Numbers JP16K01408 and JP17H04696.

**Competing interests:** The SoftBank Robotics Group Corp. provided support in the form of salaries for authors [N. M. and K. S.], but did not have any additional role in the study design, data collection and analysis, decision to publish, or preparation of the manuscript. The specific roles of these authors have been articulated in the "author contributions" section. The authors [N. M. and K. S.] are employed by SoftBank Robotics Group Corp. This does not alter our adherence to PLOS ONE policies on sharing data and materials.

cues (observation data), such as facial expressions and voice tones. The unknown emotions correspond to the hypotheses in Bayesian inference; i.e., it is the inferred target. In addition, a probability distribution representing what type of facial expression appears in what proportion when the other has a specified emotion corresponds to a model for each hypothesis. For example, if one has a model that "If a person is pleased, the person will smile, with an 80% chance," and if one observes that person to smile frequently, they will be more confident in the hypothesis that "the person is pleased." That is, by observing the data, the "effect" of a "smile," the emotion of "joy" is presumed as the cause of the "smile."

Attention should be paid to the following two points. First, to infer someone's emotions more accurately, it is better to have as much observation data as possible, but only if it is ensured that the emotion will not change during the estimation. Emotions change from moment to moment. In such an unsteady situation, it is necessary to consider whether the observation data are derived from the same emotion throughout the period being observed for the estimation. This may be a problem with online clustering of non-stationary data. The second point is that because a model for a stranger cannot be given in advance, it is necessary to learn and construct it from observed data. If this model is wrong, one cannot obtain the correct results from observations to determine the emotion of the person.

Regarding the second point, there are methods such as the expectation-maximization (EM) algorithm and the K-means algorithm that perform inference and learning simultaneously. The EM algorithm is a method for obtaining the maximum likelihood estimate in a hidden-variable model [3] and it is often used for mixture models or latent-topic models, such as latent Dirichlet allocation [4]. K-means is the non-stochastic version of the EM algorithm [3].

In the previous example, hidden variables correspond to emotions such as joy or anger because they are not observable directly. By using the EM algorithm, a person's emotions can be estimated from the observed data while creating a model of emotion, based on the observed data. However, in the EM algorithm, it is necessary to provide all the observational data at one time. In practice, there are cases where data processing must be performed sequentially after each time the data are observed.

Various online algorithms have been proposed to deal with this situation [5–13]. For example, Yamanishi et al. [7] proposed a sequential discounting expectation-maximization (SDEM) algorithm that introduced the effect of forgetting to deal with unsteady situations where the inferred target changed. The algorithm is used in the fields of anomaly detection and change point detection. We also proposed EBI, incorporating causal reasoning into Bayesian inference, like an algorithm that performs inference and learning simultaneously [14].

In the field of cognitive psychology, experiments on causal induction have been performed to identify how humans evaluate the strength of causal relations between two events [15–19]. In a regular conditional statement of the form "if p then q," the degree of confidence is considered to be proportional to the conditional probability $P(q|p)$, which is the probability of occurrence of $q$ given the existence of $p$ [20]. In contrast, in the case of a causal relation, it has been experimentally demonstrated that humans have a strong sense of causal relation between a cause $c$ and an effect $e$ when $P(c|e)$ is high, as well as when $P(e|c)$ is high. Specifically, the causal intensity that people feel between $c$ and $e$ can be approximated by the geometric mean of $P(e|c)$ and $P(c|e)$. This is called the "dual-factor heuristics" (DFH) model [19]. If the causal intensity between $c$ and $e$ is denoted as $DFH(e|c)$, then $DFH(e|c) = \sqrt{P(e|c)P(c|e)} \neq P(e|c)$. Here, note that $DFH(c|e) = DFH(e|c)$ is valid. Such inference is called "symmetry inference."

In this paper, we first describe the EBI, which replaces conditional inference in Bayesian inference with causal inference. Second, we show that the learning effect and forgetting effect are introduced into Bayesian inference by this replacement. Third, we evaluate the estimation

performance of the EBI through the learning task of a dynamically changing Gaussian mixture model. In the evaluation, the performance is compared with SDEM, an online EM algorithm.

## Methods

### Bayesian inference

In Bayesian inference, several hypotheses $h_k$ are first defined and models (probability distributions of data $d$ for the hypotheses) are prepared in the form of conditional probabilities $P(d|h_k)$. This conditional probability is called "likelihood" in the case that the data are fixed, and this probability is considered to be a function of the hypothesis. In addition, the confidence $P(h_k)$ for each hypothesis is prepared as a prior probability. That is, one must have some prior estimate of the probability that $h_k$ is true.

Assuming that the confidence for hypothesis $h_k$ at time $t$ is represented as $P^t(h_k)$ and data $d^t$ were observed, the posterior probability is calculated as follows using Bayes' theorem.

$$P^t(h_k|d^t) = \frac{P^t(h_k)P(d^t|h_k)}{P^t(d^t)} \tag{1}$$

where $P^t(d^t)$ is the marginal probability of $d^t$ at time $t$ and is defined as follows.

$$P^t(d^t) = \sum_k P^t(h_k)P(d^t|h_k) \tag{2}$$

Next, the posterior probability that resulted from the analysis becomes the new prior estimate in the update.

$$P^{t+1}(h_k) \leftarrow P^t(h_k|d^t) \tag{3}$$

By combining formulas (1) and (3), we can get formula (4).

$$P^{t+1}(h_k) \leftarrow \frac{P^t(h_k)P(d^t|h_k)}{P^t(d^t)} \tag{4}$$

Each time the data are observed, the inference progresses by updating the confidence for each hypothesis using formula (4). Note that in this process, the confidence $P^t(h_k)$ for each hypothesis changes over time, but the model $P(d|h_k)$ for each hypothesis does not change.

If we focus on the recursiveness of $P^t(h_k)$, formula (4) can be rewritten as

$$P^{t+1}(h_k) \leftarrow P^1(h_k)\prod_{i=1}^{t}\frac{P(d^i|h_k)}{P^i(d^i)} \tag{5}$$

Here, the denominator $P^i(d^i)$ is common to all hypotheses and can be considered as a constant. Therefore, if normalization processing is omitted, formula (5) can be written as follows.

$$P^{t+1}(h_k) \leftarrow P^1(h_k)\prod_{i=1}^{t}P(d^i|h_k) \tag{6}$$

That is, the current confidence for a hypothesis is proportional to the prior probability multiplied by the likelihood of the data observed so far.

### Extended Bayesian inference

We proposed the incorporation of such causal induction factor into Bayesian inference in the EBI [14]. In the EBI, first, we defined $C(e|c)$ as a new index representing the strength of the

connection between two events $c$ and $e$ as follows.

$$C(e|c) = [(1-\alpha)P(e|c)^m + \alpha P(c|e)^m]^{1/m} \tag{7}$$

Similarly, we defined $C(c|e)$ as follows.

$$C(c|e) = [(1-\alpha)P(c|e)^m + \alpha P(e|c)^m]^{1/m} \tag{8}$$

These formulas represent the generalized weighted averages of $P(e|c)$ and $P(c|e)$. The generalized weighted average of variables $x$ and $y$ is expressed by the following formula using parameters $\alpha$ and $m$.

$$\mu(x,y|\alpha,m) = [(1-\alpha)x^m + \alpha y^m]^{1/m} \tag{9}$$

Here, $\alpha$ takes values in the range $0 \leq \alpha \leq 1$ and denotes weighting the values of $x$ and $y$, while $m$ takes values in the range $-\infty \leq m \leq \infty$ and denotes the manner of taking the mean. For example, suppose $\alpha = 0.5$ and $m = 1$, then $\mu(x,y|0.5,1) = 0.5x+0.5y$, which represents the arithmetic mean. Suppose $\alpha = 0.5$ and $m = -1$, then $\mu(x,y|0.5,-1) = 2xy/(x+y)$ represents the harmonic mean.

Suppose that $m = 0$, then formula (9) is undefinable. If $X \approx 0$, the approximation of $\exp(X) \approx 1+X$ is established by the Maclaurin expansion. When $x$ is finite and $m \approx 0$, the approximation $m\log(x) \approx 0$ holds true; thus, the approximation $x^m = \exp(\log(x^m)) = \exp(m\log(x)) \approx 1+m\log(x)$ is derived. Similarly, when $y$ is finite and $m \approx 0$, $y^m \approx 1+m\log(y)$ is derived. Therefore, formula (9) can be rewritten as follows.

$$
\begin{aligned}
\mu(x,y|\alpha,m) &= [(1-\alpha)x^m + \alpha y^m]^{1/m} \\
&\approx [(1-\alpha)(1+m\log x) + \alpha(1+m\log y)]^{1/m} \\
&= [1 + m\log x^{1-\alpha} + m\log y^\alpha]^{1/m} \\
&= [1 + m\log x^{1-\alpha}y^\alpha]^{1/m} \\
&= [1 + \log(x^{1-\alpha}y^\alpha)^m]^{1/m} \\
&\approx [\exp(\log(x^{1-\alpha}y^\alpha)^m)]^{1/m} \\
&= \exp(\log(x^{1-\alpha}y^\alpha)) \\
&= x^{1-\alpha}y^\alpha
\end{aligned}
\tag{10}
$$

Therefore, if we define the mean value $\mu(x,y|\alpha,0)$ as the limit of $m \rightarrow 0$, we get $\mu(x,y|\alpha,0) = x^{1-\alpha}y^\alpha$, where if $\alpha = 0.5$, it denotes the geometric mean $\mu(x,y|0.5,0) = \sqrt{xy}$. That is, if $m = 0$ and $\alpha = 0.5$, $C(e|c) = DFH(e|c) = \sqrt{P(e|c)P(c|e)}$. In contrast, if $\alpha = 0$, $C(e|c) = P(e|c)$, irrespective of the value of $m$. In other words, by introducing parameters $m$ and $\alpha$, the conditional reasoning $P(e|c)$ and causal reasoning $DFH(e|c)$ can be seamlessly connected.

Here, we discuss the meaning of $m$. For simplicity, let $\alpha = 0.5, x+y = 1$ $(0 \leq x \leq 1)$.

$$\mu(x|0.5,m) = [0.5x^m + 0.5(1-x)^m]^{1/m} \tag{11}$$

When $m = 1$, $\mu(x|0.5,1) = 0.5$, irrespective of the value of $x$. When $m > 1$, $\mu(x|0.5,m)$ is a convex function and the values at both ends, $\mu(0|0.5,m)$ and $\mu(1|0.5,m)$, approach 1 as $m$ increases.

However, when $.0 < m < 1.$, $\mu(x|0.5,m)$ becomes a concave function and as $m$ approaches 0, the values $\mu(0|0.5,m)$ and $\mu(1|0.5,m)$ at both ends approach 0.

When $m<0$, the values at both ends cannot be defined, but as $x$ approaches 0 or 1, $\mu(x|0.5, m)$ approaches 0. In other words, in the range of $m \leq 0$, if either $x$ or $y$ approaches 0, their mean also approaches 0.

Next, using Bayes' theorem, formulas (7) and (8) can be transformed as follows [14]:

$$
\begin{aligned}
C(e|c) &= [(1-\alpha)P(e|c)^m + \alpha P(c|e)^m]^{1/m} \\
&= \left[(1-\alpha)\left(\frac{P(c)P(e|c)}{P(c)}\right)^m + \alpha\left(\frac{P(e)P(c|e)}{P(e)}\right)^m\right]^{1/m} \\
&= \left[(1-\alpha)\left(\frac{1}{P(c)}\right)^m + \alpha\left(\frac{1}{P(e)}\right)^m\right]^{1/m} P(c)P(e|c) \qquad (12) \\
&= [(1-\alpha)P(c)^{-m} + \alpha P(e)^{-m}]^{1/m} P(c)P(e|c) \\
&= \frac{P(c)P(e|c)}{[(1-\alpha)(P(c))^{-m} + \alpha(P(e))^{-m}]^{-1/m}}
\end{aligned}
$$

$$
\begin{aligned}
C(c|e) &= [(1-\alpha)P(c|e)^m + \alpha P(e|c)^m]^{1/m} \\
&= \frac{P(c)P(e|c)}{[(1-\alpha)P(e)^{-m} + \alpha P(c)^{-m}]^{-1/m}} \qquad (13)
\end{aligned}
$$

Note that $P(c)P(e|c) = P(e)P(c|e) = P(c,e)$.

Here, if we describe formulas (12) and (13) recursively, and replace $c$ and $e$ with $h_k$ and $d^t$, respectively, we can get the next formulas.

$$
C^{t+1}(d^t|h_k) \leftarrow \frac{C^t(h_k)C^t(d^t|h_k)}{[(1-\alpha)C^t(h_k)^{-m} + \alpha C^t(d^t)^{-m}]^{-1/m}} \qquad (14)
$$

$$
C^{t+1}(h_k|d^t) \leftarrow \frac{C^t(h_k)C^t(d^t|h_k)}{[(1-\alpha)C^t(d^t)^{-m} + \alpha C^t(h_k)^{-m}]^{-1/m}} \qquad (15)
$$

$$
C^t(d^t) = \sum_k C^t(h_k)C^t(d^t|h_k) \qquad (16)
$$

Formula (15) can be rewritten as follows by using a Bayesian update.

$$
C^{t+1}(h_k) \leftarrow \frac{C^t(h_k)C^t(d^t|h_k)}{[(1-\alpha)C^t(d^t)^{-m} + \alpha C^t(h_k)^{-m}]^{-1/m}} \qquad (17)
$$

In formula (17), a description of the normalization process for setting the confidence as a probability is omitted.

Assuming $\alpha = 0$ in formula (17), the same form as formula (4) for Bayesian inference is obtained.

$$
C^{t+1}(h_k) \leftarrow \frac{C^t(h_k)C^t(d^t|h_k)}{C^t(d^t)} \qquad (18)
$$

If $\alpha = 0$, $C^t(d^t|h_k)$ does not change by formula (14), as shown below.

$$
C^{t+1}(d^t|h_k) \leftarrow C^t(d^t|h_k) \qquad (19)
$$

In other words, if $\alpha = 0$, then formula (14) substantially disappears and the EBI becomes the same as Bayesian inference. In contrast, in the case of $\alpha > 0$, the likelihood is modified by

formula ([14]). In this study, we only update the likelihood of the hypothesis with the highest confidence at that time instead of updating the likelihood of all hypotheses. That is, the following formula is used instead of formula ([14]).

$$C^{t+1}(d^t|h_k) \leftarrow \begin{cases} \dfrac{C^t(h_k)C^t(d^t|h_k)}{[(1-\alpha)C^t(h_k)^{-m} + \alpha C^t(d^t)^{-m}]^{-1/m}} & \text{if } h_k = \arg\max_{h_i} C^t(h_i) \\ C^t(d^t|h_k) & \text{otherwise} \end{cases} \quad (20)$$

Hereafter, the hypothesis with the highest confidence at time $t$ is denoted by $h_{\max}^t$. If there are multiple hypotheses with the highest confidence, one of them is selected at random.

In the following, let us analyze the case of $m = 0$, that is, the geometric mean case. In the case of $m = 0$, formula ([17]) can be transformed as follows.

$$C^{t+1}(h_k) \leftarrow \frac{C^t(h_k)C^t(d^t|h_k)}{C^t(d^t)^{1-\alpha}C^t(h_k)^{\alpha}} = \left[\frac{C^t(h_k)}{C^t(d^t)}\right]^{1-\alpha} C^t(d^t|h_k) \quad (21)$$

If we focus on the recursiveness of $C^t(h_k)$, formula ([21]) can be rewritten as follows.

$$C^{t+1}(h_k) \leftarrow [C^1(h_k)]^{(1-\alpha)^t} \prod_{i=1}^{t} \frac{[C^i(d^i|h_k)]^{(1-\alpha)^{t-i}}}{[C^i(d^i)]^{(1-\alpha)^{t+1-i}}} \quad (22)$$

Here, the denominator $C^i(d^i)$ is common to all hypotheses and can be considered as a constant. Therefore, if the normalization processing is omitted, formula ([20]) can be written as follows.

$$C^{t+1}(h_k) \leftarrow [C^1(h_k)]^{(1-\alpha)^t} \prod_{i=1}^{t} [C^i(d^i|h_k)]^{(1-\alpha)^{t-i}} \quad (23)$$

This can be understood as indication that the current confidence for a hypothesis is proportional to its prior probability multiplied by the likelihood designed to weaken the weight of the distant past. In the case of $\alpha = 0$, that is, Bayesian inference, $C^{t+1}(h_k) \leftarrow C^1(h_k)C^1(d^1|h_k)C^2(d^2|h_k)\cdots C^t(d^t|h_k)$. This means that the current likelihood and the past likelihood are weighted equally.

However, in the case of $\alpha = 1$, $C^{t+1}(h_k) \leftarrow C^t(d^t|h_k)$. It means that the confidence is calculated using only the current likelihood. Thus, it can be said that the EBI introduces the effect of forgetting into Bayesian inference when considering past history.

In the case of $m = 0$, with respect to $h_{\max}^t$, formula ([20]) can be written as follows.

$$C^{t+1}(d^t|h_{\max}^t) \leftarrow \frac{C^t(h_{\max}^t)C^t(d^t|h_{\max}^t)}{C^t(h_{\max}^t)^{1-\alpha}C^t(d^t)^{\alpha}} = \left[\frac{C^t(h_{\max}^t)}{C^t(d^t)}\right]^{\alpha} C^t(d^t|h_{\max}^t) \quad (24)$$

In the case of $C^t(h_{\max}^t) > C^t(d^t)$, the likelihood becomes larger, and in the case of $C^t(h_{\max}^t) < C^t(d^t)$, the likelihood becomes smaller. This means that the model is corrected based on the observed data. Thus, it can be said that the EBI introduces the effect of learning into the Bayesian inference.

## Testing a normal distribution model

**Mean value estimation using normal distribution.** In this study, we deal with one-dimensional continuous probability distribution, such as one-dimensional normal distribution as a concrete model for the hypothesis. The model of hypothesis $k$ at time $t$ is denoted by $F(d|\theta_k^t)$. Here, $\theta_k^t$ represents the parameter of the model. In the case of normal distribution,

$F(d|\theta_k^t) = N(d|\theta_k^t)$, $\theta_k^t = (\mu_k^t, \Sigma_k^t)$, where $\mu$ and $\Sigma$ represent the mean and variance, respectively. This section describes the mean value estimation. The variance estimation will be described in the next section. When a normal distribution is used as a model, $C^t(d|h_k)$ and $C^t(d)$ are probability densities, while $C^t(h_k)$ is a probability because the number of hypotheses is discrete and finite. Thus, when calculating formulas (23) and (24), a positive number $\Delta$ is introduced and approximately calculated as follows.

$$C^{t+1}(h_k) \leftarrow [C^t(h_k)]^{1-\alpha} \Delta F(d^t|\theta_k^t) \tag{25}$$

$$C^{t+1}(d^t|h_{\max}^t) \leftarrow \frac{1}{\Delta} \left[ \frac{C^t(h_{\max}^t)}{\sum_k C^t(h_k) \Delta F(d^t|\theta_k^t)} \right]^\alpha \Delta F(d^t|\theta_{\max}^t)$$
$$= \frac{1}{\Delta^\alpha} \left[ \frac{C^t(h_{\max}^t)}{\sum_k C^t(h_k) F(d^t|\theta_k^t)} \right]^\alpha F(d^t|\theta_{\max}^t) \tag{26}$$

Here, $\theta_{\max}^t$ represents the parameter of the distribution that is the model for $h_{\max}^t$.

In formula (25), the term $\Delta$ is common to all hypotheses and can be canceled by normalization. Thus, if normalization processing is omitted, it can be expressed as follows.

$$C^{t+1}(h_k) \leftarrow [C^t(h_k)]^{1-\alpha} F(d^t|\theta_k^t) \tag{27}$$

In formula (27), if the confidence for a hypothesis becomes zero once, it remains zero thereafter. To prevent this, normalization processing (smoothing) is performed by adding a small positive constant $\varepsilon$ to the confidence of each hypothesis obtained by formula (27).

$$C^{t+1}(h_k) \leftarrow \frac{C^{t+1}(h_k) + \varepsilon}{\sum_{j=1}^{K} [C^{t+1}(h_j) + \varepsilon]} = \frac{C^{t+1}(h_k) + \varepsilon}{K\varepsilon + \sum_{j=1}^{K} C^{t+1}(h_j)} \tag{28}$$

Here, $K$ represents the total number of hypotheses. In this study, we set $\varepsilon = 10^{-10}$.

Having observed the data $d^t$, the likelihood is changed to $C^{t+1}(d^t|h_{\max}^t)$ by formula (26). Concomitantly, the parameter of the model for the hypothesis is modified from $\theta_{\max}^t$ to $\theta_{\max}^{t+1}$ so that the following equation is satisfied.

$$C^{t+1}(d^t|h_{\max}^t) = F(d^t|\theta_{\max}^{t+1}) \tag{29}$$

If $F$ is a normal distribution, Eq (29) can be described as follows.

$$C^{t+1}(d^t|h_{\max}^t) = F(d^t|\theta_{\max}^{t+1}) = \frac{1}{\sqrt{2\pi\Sigma_{\max}^{t+1}}} \exp\left[ -\frac{(d^t - \mu_{\max}^{t+1})^2}{2\Sigma_{\max}^{t+1}} \right] \tag{30}$$

Updating the variance from $\Sigma_{\max}^t$ to $\Sigma_{\max}^{t+1}$ is described in the next section.
Solving formula (30) for $\mu_{\max}^{t+1}$ leads to the following two solutions.

$$\mu_1 = d^t + \sqrt{-2\Sigma^t \log[C^{t+1}(d^t|h_{\max}^t)\sqrt{2\pi\Sigma_{\max}^{t+1}}]}$$
$$\mu_2 = d^t - \sqrt{-2\Sigma^t \log[C^{t+1}(d^t|h_{\max}^t)\sqrt{2\pi\Sigma_{\max}^{t+1}}]} \tag{31}$$

$\mu_{\max}^t$ reflects the past observed data. We determine $\mu_{\max}^{t+1}$ as the one closer to $\mu_{\max}^t$, among the

two solutions $\mu_1$ and $\mu_2$ to account for the past data as much as possible.

$$
\mu_{max}^{t+1} = \begin{cases} \mu_1 & if \quad |\mu_1 - \mu_{max}^t| \leq |\mu_2 - \mu_{max}^t| \\ \mu_2 & otherwise \end{cases} \tag{32}
$$

However, to solve formula (29), $C^{t+1}(d^t|h_{max}^t)$ needs to be within the range of $0 < C^{t+1}(d^t|h_{max}^t) \leq \frac{1}{\sqrt{2\pi\Sigma_{max}^{t+1}}}$. Thus, we set the following restrictions after calculating $C^{t+1}(d^t|h_{max}^t)$ using formula (26).

$$
C^{t+1}(d^t|h_{max}^t) \leftarrow \min\left(\max(C^{t+1}(d^t|h_{max}^t), \varepsilon), \frac{1}{\sqrt{2\pi\Sigma_{max}^{t+1}}}\right) \tag{33}
$$

where $\max(x,y)$ represents the larger of $x$ and $y$. Conversely, $\min(x,y)$ represents the smaller of $x$ and $y$. We set $\varepsilon = 10^{-10}$.

If $K = 1$, there is no other hypothesis. Therefore, the only hypothesis always becomes $h_{max}^t$ and the value of confidence is always 1. Consider the situation $C^t(h_{max}^t) \approx 1$, including the case of $K = 1$. In this case, the confidence of hypotheses other than $h_{max}^t$ becomes almost 0 by the constraint $\sum_k C^t(h_k) = 1$; thus, $C^t(d^t) = \sum_k C^t(h_k)C^t(d^t|h_k) \approx C^t(d^t|h_{max}^t)$ is derived from formula (16). Therefore, formula (26) can be transformed as follows.

$$
\begin{aligned}
C^{t+1}(d^t|h_{max}^t) &\leftarrow \frac{1}{\Delta^\alpha}\left[\frac{C^t(h_{max}^t)}{\sum_k C^t(h_k)N(d^t|\mu_k^t, \Sigma_k^t)}\right]^\alpha N(d^t|\mu_{max}^t, \Sigma_{max}^t) \\
&\approx \left(\frac{1}{\Delta}\right)^\alpha [C^t(d^t|h_{max}^t)]^{1-\alpha}
\end{aligned} \tag{34}
$$

If formula (34) is denoted by $x^{t+1} = f(x^t) = \left(\frac{1}{\Delta}\right)^\alpha (x^t)^{1-\alpha}$, $f(x^t)$ becomes a concave function. Solving $x^t = f(x^t)$ results in $x^t = 0, \frac{1}{\Delta}$. The fixed point $(x^t, f(x^t)) = \left(\frac{1}{\Delta}, \frac{1}{\Delta}\right)$ is a stable point because $x^t \geq f(x^t)$ when $x^t > \frac{1}{\Delta}$ and $x^t \leq f(x^t)$ when $x^t < \frac{1}{\Delta}$. In this study, we set $\Delta = \sqrt{2\pi\Sigma_{max}^t}$. In this case, $C^{t+1}(d^t|h_{max}^t)$ approaches the vertex of the normal distribution whenever data $d^t$ are observed.

As shown in formula (29), $\mu_{max}^{t+1}$ is determined to satisfy the condition $C^{t+1}(d^t|h_{max}^t) = N(d^t|\mu_{max}^{t+1}, \Sigma_{max}^{t+1})$. This means that $\mu_{max}^{t+1}$ approaches the observation data $d^t$. Through the processing described above, the confidences for each hypothesis and the model for the hypothesis with maximum confidence are corrected whenever the data are observed.

We will hereinafter refer to the latter process of modifying the model for $h_{max}^t$ as inverse Bayesian inference [21–24]. If the former process of updating the confidences for hypotheses is referred to as inference, inverse Bayesian inference can be called "learning" because it forms a model for a hypothetical instead of an inference. Thus, although the two $\alpha$ s in formulas (26) and (27) are denoted by the same $\alpha$, they can be called the "learning rate" and "forgetting rate," respectively. We can also set the "learning rate" and "forgetting rate" as two independent parameters. However, when dealing with temporal alteration, like in this study, good performance is achieved when the two parameters have almost identical values. On the contrary, it is preferable to set the parameters separately in spatial clustering.

**Variance estimation using gamma distribution.** Consider a random variable $D$ that is the sum of $n$ squares of data sampled from a normal distribution $N(0,\Sigma)$ with mean 0 and

variance $\Sigma$.

$$d^i \sim N(0, \Sigma), \ D = \sum_{i=1}^{n}(d^i)^2 \tag{35}$$

In this case, $D$ follows the gamma distribution with shape parameter $S = n/2$ and scale parameter $\lambda = 1/(2\Sigma)$, as shown below.

$$f(D|\lambda, S) = \frac{\lambda^S}{\Gamma(S)}D^{S-1}\exp(-\lambda D) = \frac{1}{\Gamma(n/2)(2\Sigma)^{n/2}}D^{n/2-1}\exp(-D/2\Sigma) \tag{36}$$

The mean of this distribution is $S/\lambda = n\Sigma$. We set $n = 20$, that is $S = 10$. We use the gamma distribution as a model for estimating variance from observed data $d^t$.

First, the following $D_k^t$ is calculated using the mean estimated value $\mu_k^t$ obtained in the previous section. This is used as input data for variance estimation instead of the observation data $d^t$.

$$T_k(n_k \% n + 1) = t, \ n_k \leftarrow n_k + 1, \quad if \ h_k^t = h_{max}^t$$

$$D_k^t = \sum_{i=1}^{n}(d^{T_k(i)} - \mu_k^{T_k(i)})^2 \tag{37}$$

Here, $n_k$ represents the number of times that hypothesis $k$ has been the hypothesis with the highest confidence. The initial value of $n_k$ is 0. $x \% y$ represents the remainder when integer $x$ is divided by integer $y$.

The model of hypothesis $k$ at time $t$ is $F(D|\theta_k^t) = f(D|\theta_k^t), \ \theta_k^t = (\lambda_k^t, S)$. In this case, formula (29) is rewritten as

$$C^{t+1}(D_{max}^t|h_{max}^t) = F(D_{max}^t|\theta_{max}^{t+1}) = f(D_{max}^t|\lambda_{max}^{t+1}, S) = \frac{(\lambda_{max}^{t+1})^S}{\Gamma(S)}(D_{max}^t)^{S-1}\exp(-\lambda_{max}^{t+1}D_{max}^t) \tag{38}$$

With $\Theta_{max}^{t+1} = -\lambda_{max}^{t+1}D_{max}^t/S, \ Z^{t+1} = -(D_{max}^t C^{t+1}(D_{max}^t|h_{max}^t)\Gamma(S))^{1/S}/S$, this equation can be rewritten as

$$Z^{t+1} = \Theta_{max}^{t+1}\exp(\Theta_{max}^{t+1}) \tag{39}$$

For this equation to have a solution with respect to $\Theta_{max}^{t+1}$ in the range of $\Theta_{max}^{t+1} < 0, -1/e \leq Z^{t+1} < 0$ must be satisfied. Therefore, the following restrictions are provided.

$$Z^{t+1} \leftarrow \max(\min(Z^{t+1}, -\varepsilon), -1/e) \tag{40}$$

We set $\varepsilon = 10^{-10}$.

When formula (39) is solved for $\Theta_{max}^{t+1}$, the following two solutions are obtained.

$$\Theta_1 = W_{-1}(Z^{t+1})$$
$$\Theta_2 = W_0(Z^{t+1}) \tag{41}$$

Here, $W_{-1}$ and $W_0$ are two Lambert W functions that satisfy $Z = xe^x \Leftrightarrow x = W(Z)$.

Similar to the case of estimating the mean value, $\Theta_{max}^{t+1}$ is determined as

$$\Theta_{max}^{t+1} = \begin{cases} \Theta_1 & if \ |\Theta_1 - \Theta_{max}^t| \leq |\Theta_2 - \Theta_{max}^t| \\ \Theta_2 & otherwise \end{cases} \tag{42}$$

Because $\Theta_{\max}^{t+1} = -\lambda_{\max}^{t+1} D_{\max}^t / S$, the scale parameter $\lambda_{\max}^{t+1}$ can be calculated as

$$\lambda_{\max}^{t+1} = -\frac{S\Theta_{\max}^{t+1}}{D_{\max}^t} \tag{43}$$

Further, because $\lambda_{\max}^{t+1} = 1/(2\Sigma_{\max}^{t+1})$, the variance estimate $\Sigma_{\max}^{t+1}$ is calculated as

$$\Sigma_{\max}^{t+1} = \frac{1}{2\lambda_{\max}^{t+1}} \tag{44}$$

We use $\Sigma_{\max}^{t+1}$ as the variance estimate for the next time in generation distribution. That is, $\theta_{\max}^{t+1} = (\mu_{\max}^{t+1}, \Sigma_{\max}^{t+1})$.

Regarding the value of $\Delta$, we consider the gamma distribution in which the current input value $D_{\max}^t$ is the mean value $S/\lambda$, and define the output value for the input value $D_{\max}^t$ as $1/\Delta$, based on the same arguments as in the case of normal distribution.

$$\Delta = \frac{1}{f(D_{\max}^t | S/D_{\max}^t, S)} = \frac{\Gamma(S)D_{\max}^t \exp(S)}{S^S} \tag{45}$$

The group of processes described in this section and the previous section is summarized as an algorithm below.

1. Set values for parameters $\alpha$, $m$, $\varepsilon$, $K$.

2. Establish initial values for $\theta_k^1 = (\mu_k^1, \Sigma_k^1)$, $C^1(h_k)$ $(k = 1, 2, \cdots K)$.

3. Repeat the following whenever data $d^t$ are observed.

   - Update the confidence $C^{t+1}(h_k)$ of each hypothesis using formulas (27) and (28).

   - Find the hypothesis $h_{\max}^t$ with the maximum confidence.

   - Create the input data $D_{\max}^t$ for variance calculation using formula (37).

   - Update the likelihood $C^{t+1}(D_{\max}^t | h_{\max}^t)$ of the hypothesis $h_{\max}^t$ for the input data $D_{\max}^t$ using formula (26).

   - Correct the variance $\Sigma_{\max}^{t+1}$ of the model for the hypothesis $h_{\max}^t$ using formulas (41), (42), (43), and (44) to match the new likelihood $C^{t+1}(D_{\max}^t | h_{\max}^t)$.

   - Update the likelihood $C^{t+1}(d^t | h_{\max}^t)$ of the hypothesis $h_{\max}^t$ for the observed data $d^t$ using formula (26).

   - Correct the mean $\mu_{\max}^{t+1}$ of the model for the hypothesis $h_{\max}^t$ using formulas (31) and (32) to match the new likelihood $C^{t+1}(d^t | h_{\max}^t)$.

   - $\theta_{\max}^{t+1} = (\mu_{\max}^{t+1}, \Sigma_{\max}^{t+1})$ is set as the new parameter of the model for the hypothesis $h_{\max}^t$.

## Sequential Discounting Expectation-Maximization Algorithm (SDEM)

This section describes SDEM, an online EM algorithm proposed by Yamanishi et al. [7]. In SDEM, the E and M steps are executed once for each data observed sequentially. First, in step E, the responsibility is calculated. The responsibility for the normal distribution $k$ of the data $d^t$

is calculated as follows.

$$q_k^t = \frac{\pi_k^t N(d^t | \mu_k^t, \Sigma_k^t)}{\sum_{j=1}^{K} \pi_j^t N(d^t | \mu_j^t, \Sigma_j^t)} \tag{46}$$

Here, $\sum_{k=1}^{K} \pi_k^t = 1$ is assumed. $\pi_k$ is called the "mixing weights" and represents the weight of each normal distribution.

Next, in the M step, the mixing weights, means, and variances of each normal distribution are updated. However, weighting is performed to weaken the influence of older observation data by introducing the discounting rate $\beta(0<\beta<1)$.

$$\tilde{\pi}_k^{t+1} \leftarrow (1 - \beta)\tilde{\pi}_k^t + \beta q_k^t = \beta \sum_{i=1}^{t} (1 - \beta)^{t-i} q_k^i \tag{47}$$

$$\tilde{\mu}_k^{t+1} \leftarrow (1 - \beta)\tilde{\mu}_k^t + \beta q_k^t d^t = \beta \sum_{i=1}^{t} (1 - \beta)^{t-i} q_k^t d^i \tag{48}$$

$$\tilde{\Sigma}_k^{t+1} \leftarrow (1 - \beta)\tilde{\Sigma}_k^t + \beta q_k^t (d^t - \mu_k^t)(d^t - \mu_k^t) = \beta \sum_{i=1}^{t} (1 - \beta)^{t-i} q_k^t (d^i - \mu_k^t)(d^i - \mu_k^t) \tag{49}$$

$$\mu_k^{t+1} \leftarrow \frac{\tilde{\mu}_k^{t+1}}{\tilde{\pi}_k^{t+1}} \tag{50}$$

$$\Sigma_k^{t+1} \leftarrow \frac{\tilde{\Sigma}_k^{t+1}}{\tilde{\pi}_k^{t+1}} \tag{51}$$

Regarding $\tilde{\pi}_k^{t+1}$, smoothing is performed to prevent it from becoming 0 and normalize, similar to the EBI.

$$\pi_k^{t+1} \leftarrow \frac{\tilde{\pi}_k^{t+1} + \gamma}{K\gamma + \sum_{j=1}^{K} \tilde{\pi}_j^{t+1}} \tag{52}$$

We set $\gamma = 0.001$ for optimal performance.

In the case of $K = 1$, $q_1^t$ and $\tilde{\pi}_1^t$ are always 1. At this time, formula (50) shows that the new estimated value is obtained as a convex combination of the current estimated value and the current observed data.

$$\mu_k^{t+1} \leftarrow (1 - \beta)\mu_k^t + \beta d^t \tag{53}$$

This represents the EMA.

By setting $\pi_k^{t+1} = P^{t+1}(h_k)$, $q_k^i = P^i(h_k|d^i)$ in formulas (47) and (52), $P^{t+1}(h_k)$ can be described approximately as follows.

$$P^{t+1}(h_k) \leftarrow \beta \sum_{i=1}^{t} (1 - \beta)^{t-i} P^i(h_k|d^i) \propto \sum_{i=1}^{t} (1 - \beta)^{t-i} P^i(h_k|d^i) \tag{54}$$

Taking the logarithm of both sides of formula (23) in the EBI, the following transformation

can be made.

$$\log(C^{t+1}(h_k)) \leftarrow \sum_{i=1}^{t} (1-\alpha)^{t-i} \log(C^i(d^i|h_k)) + (1-\alpha)^t \log(C^1(h_k)) \tag{55}$$

Comparing formula (55) with formula (54), the EBI differs in that it takes a logarithm and uses likelihood instead of posterior probability.

The group of processes described above is summarized as an algorithm below.

1. Set values for parameters $\beta$, $\gamma$, $K$.

2. Establish initial values for the variables $\theta_k^1 = (\mu_k^1, \Sigma_k^1), \pi_k^1 \;\; (k = 1, 2, \cdots K)$.

3. Repeat the following whenever data $d^t$ are observed.

   - E step: Calculate the responsibility $q_k^t$ for each normal distribution $k$ of the observed data $d^t$ using formula (46).

   - M step: Update the mixing weights $\pi_k^{t+1}$ using formulas (47) and (52).

   - M step: Correct the mean $\mu_k^{t+1}$ and variance $\Sigma_k^{t+1}$ of the normal distribution using formulas (48), (49), (50), and (51).

   - $\theta_k^{t+1} = (\mu_k^{t+1}, \Sigma_k^{t+1})$ is set as the new parameter of the model for each hypothesis $h_k^t$.

## Simulation

To investigate the behavior of EBI, a simulation was performed. In the simulation, one random number $d^t$ is generated at each time from a certain normal distribution (the "generation distribution"). Then, the EBI estimates the generation distribution by observing $d^t$.

In this study, we deal with a task in which the mean and variance of the generation distribution fluctuate randomly at each regular interval. Specifically, every 1000 steps, a random number from a uniform distribution of the range [0, 5] is generated, and the number is set as a new mean of distribution. Similarly, a random number from a uniform distribution of the range [0, 0.1] is generated, and the number is set as a new variance of the distribution.

$$\mu_{correct}^{t+1} = \begin{cases} 5 \times rnd^t & \textit{if } t \% 1000 = 0 \\ \mu_{correct}^t & \textit{otherwise} \end{cases} \tag{56}$$

$$\Sigma_{correct}^{t+1} = \begin{cases} 0.1 \times rnd^t & \textit{if } t \% 1000 = 0 \\ \Sigma_{correct}^t & \textit{otherwise} \end{cases} \tag{57}$$

Here, $\mu_{correct}^t$ and $\Sigma_{correct}^t$ represent the mean and variance of the normal distribution used as the generation distribution at time $t$, that is, the correct values in this task.

$rnd^t$ represents a random number generated from a continuous uniform distribution of the range [0, 1] at time $t$. Fig 1 shows an example of time evolution of observation data $d^t$ in this task.

In the simulation, estimations were performed via the EBI, SDEM, and EMA for comparison. The parameter estimated by the EBI at time $t$ is that of the model for $h_{max}^t$, that is $\theta_{max}^t = (\mu_{max}^t, \Sigma_{max}^t)$. The parameter estimated by SDEM is that of the distribution for which the

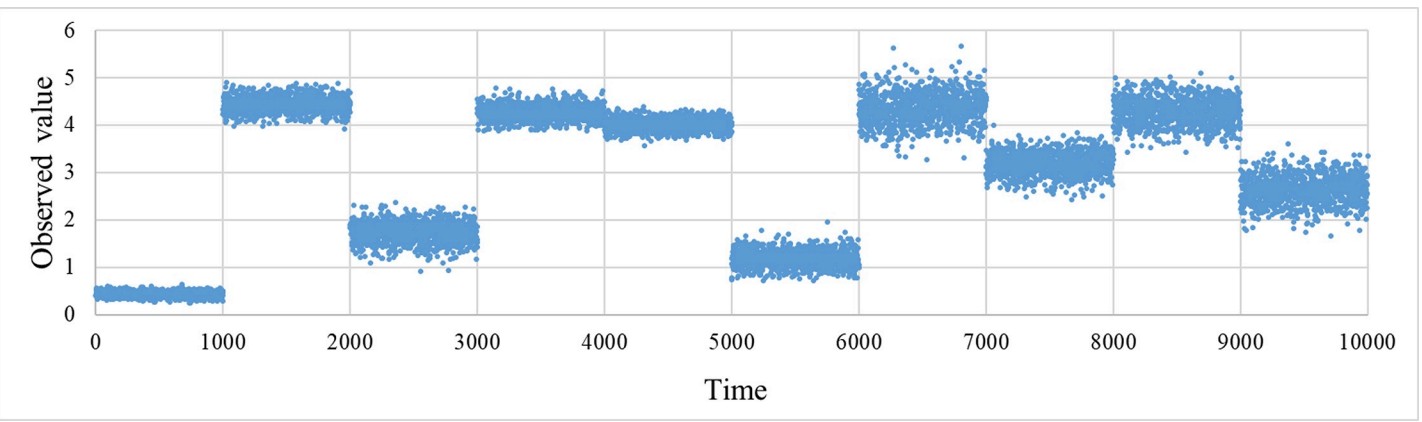

**Fig 1. Time progress of observed values.**

observed data $d^t$ have the highest responsibility among the normal distributions included in the Gaussian mixture distribution, that is, $\theta_{m^t}^t = (\mu_{m^t}^t, \Sigma_{m^t}^t)$, $m^t = \arg\max_k q_k^t$.

The mean $\mu_{EMA}^t$ and variance $\Sigma_{EMA}^t$ for the EMA are updated as follows.

$$\mu_{EMA}^{t+1} \leftarrow (1-\beta)\mu_{EMA}^t + \beta d^t \tag{58}$$

$$\Sigma_{EMA}^{t+1} \leftarrow (1-\beta)\Sigma_{EMA}^t + \beta(d^t - \mu_{EMA}^t)(d^t - \mu_{EMA}^t) \tag{59}$$

Here, β (0<β<1) represents a discount rate.

## Results

Fig 2(A) shows an example of the result by EBI. In this simulation, the initial values of the mean and variance of the model for each hypothesis were set to $\mu_k^1 = 2.5$ and $\Sigma_k^1 = 0.05$, respectively.

Fig 2(A) shows the time evolution of the correct value $\mu_{correct}^t$ and that of the estimated result by the EBI, set to $K = 10$. Fig 2(B) shows the result obtained by the EBI set to $K = 1$ (i.e., the result for inverse Bayesian inference). Fig 2(C) shows the estimation results obtained by three types of EMA with different discounting rates β. It is evident that for larger discounting rates, the responses to sudden changes are quicker, as expected, but the fluctuations are increased during the stable period.

In the case of EBI, initially, it takes time to follow up when the correct value suddenly changes. However, there are cases where changes can be handled instantly over time. In contrast, in the cases of inverse Bayesian inference and EMA, the follow-up performances are not improved over time at all.

Fig 3(A) shows the time evolution of the means $\mu_k^t$ of the models for ten hypotheses used in the simulation of Fig 2(A). Fig 3(B) shows the time evolution of the hypothesis $h_{max}^t$ with the maximum confidence. Initially, all hypothesis models are the same, but various hypothesis models are formed by learning over time. Additionally, it is evident that it is possible to follow quickly by appropriately switching the hypotheses. When the EBI is set to $K = 1$ and EMA, because the hypotheses cannot be switched, such quick tracking cannot be achieved.

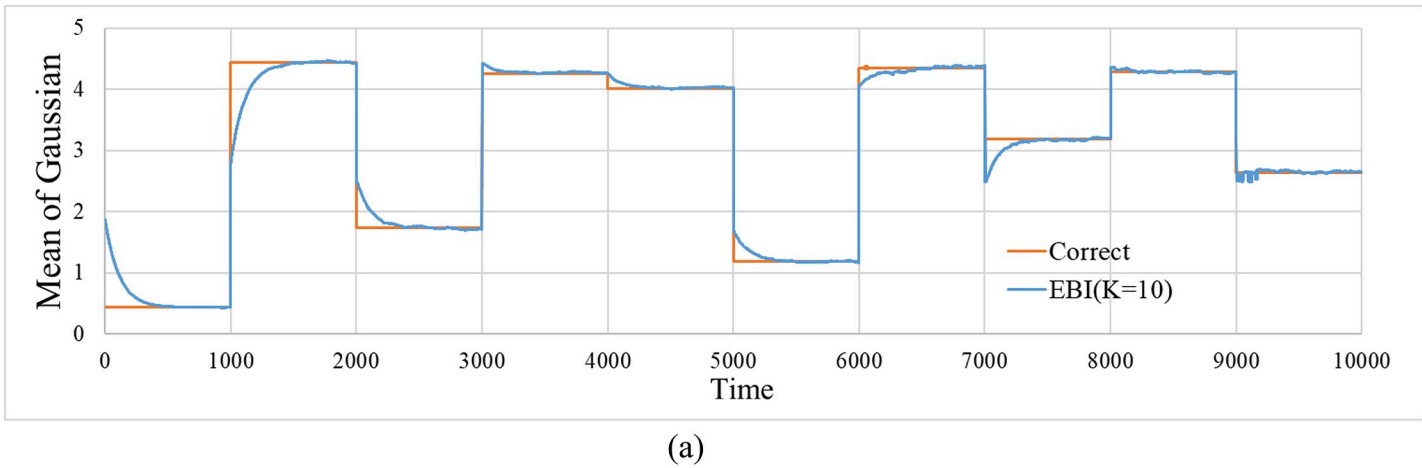

(a)

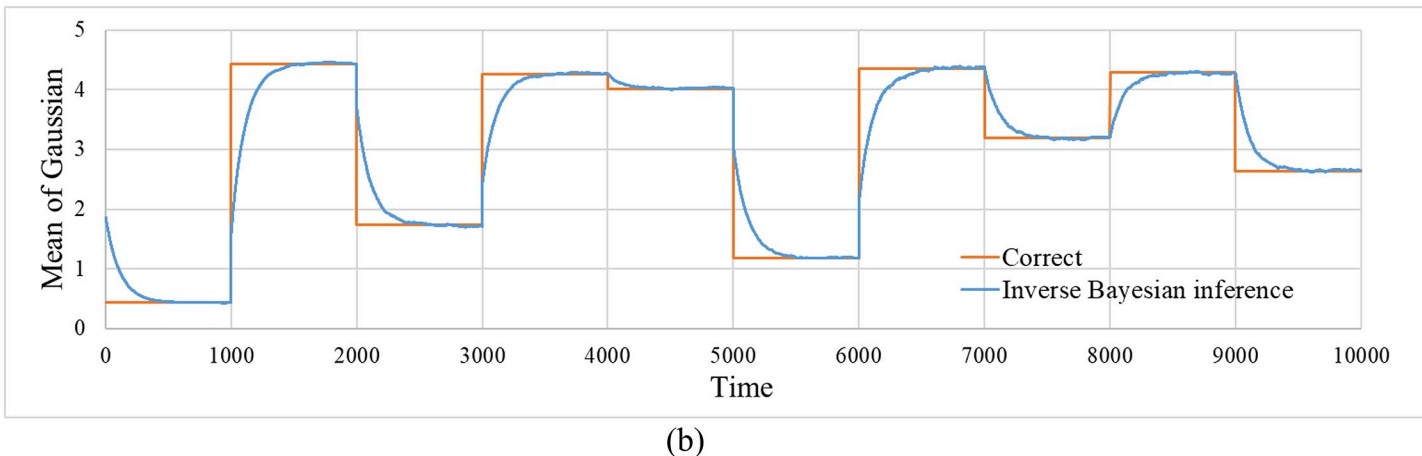

(b)

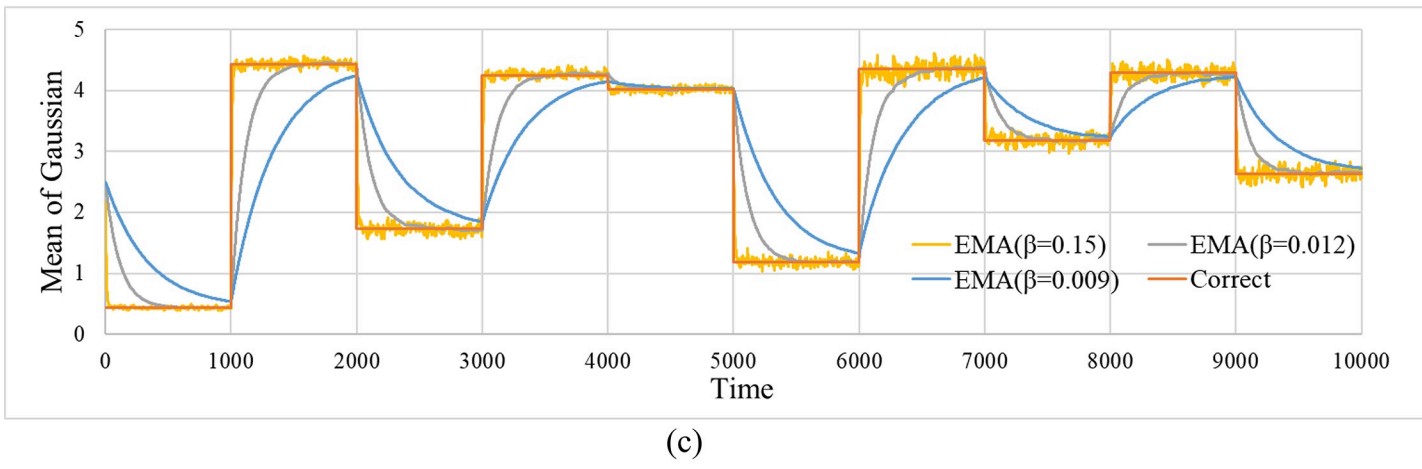

(c)

**Fig 2. Time progress of the estimated values for the mean of Gaussian.** The figures include the correct mean. (a) Estimated values by EBI. $(\alpha, m, K) = (0.018, 0.0, 10)$. (b) Estimated values by inverse Bayesian inference. $(\alpha, m, K) = (0.018, 0.0, 1)$. (c) Estimated values by EMA. $\beta = 0.009, 0.012, 0.15$.

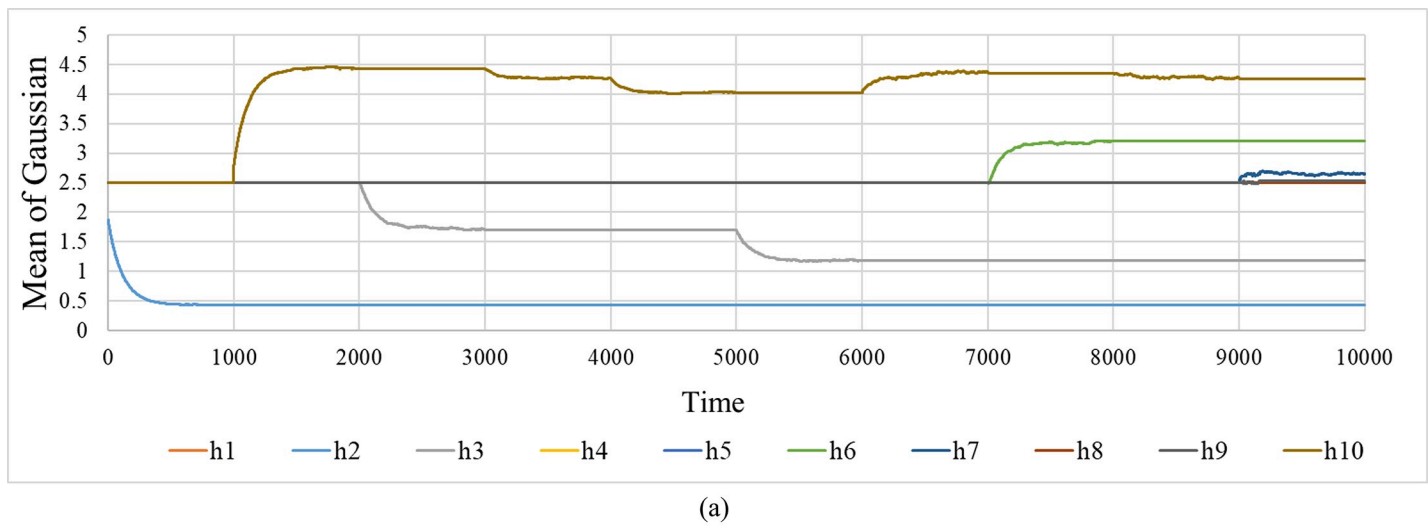

(a)

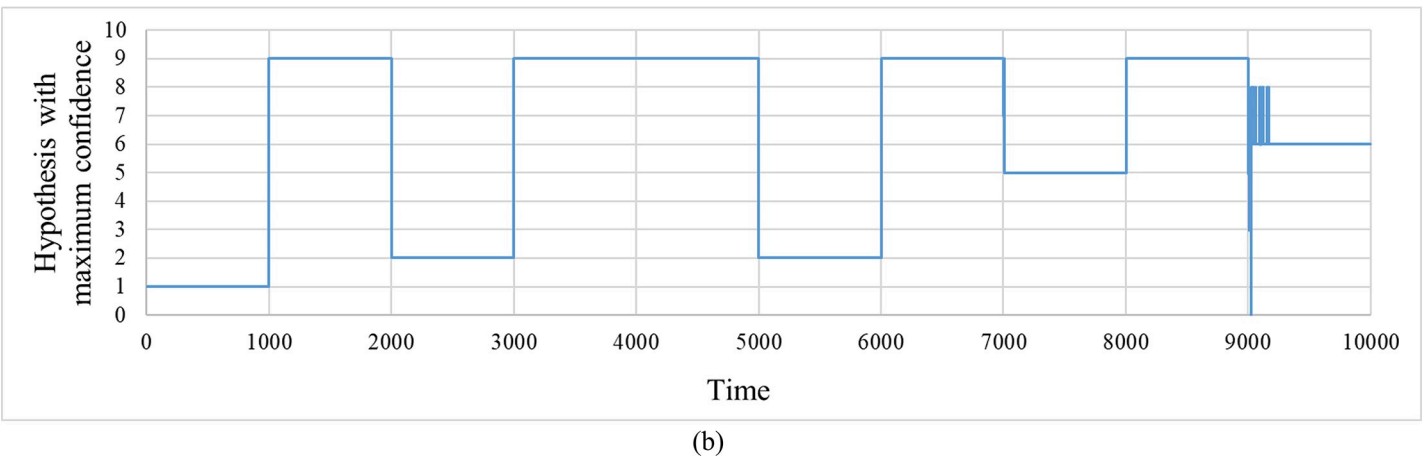

(b)

**Fig 3. Internal state of extended Bayesian inference.** (a) Time progress of the mean of normal distribution for each hypothesis. (b) Time progress of the hypothesis with the greatest degree of confidence.

To evaluate the estimation performance of each method, the root-mean-square error (RMSE) between the estimated value and correct value is calculated as follows.

$$RMSE = \sqrt{\frac{\sum_{t=j}^{T+j-1} \left( \hat{\mu}^t - \mu_{correct}^t \right)^2}{T}}$$

(60)

Here, $\hat{\mu}^t$ and $\mu_{correct}^t$ represent the estimated value and correct value at time $t$, respectively. $T$ represents a period for evaluation.

Each interval of 1000 steps, from a change in the generation distribution to the next change, is divided into two halves. We use the RMSE of the first half as a measure of the inability to follow rapid changes and that of the second half as a measure of the inaccuracy of the estimation in the stable period.

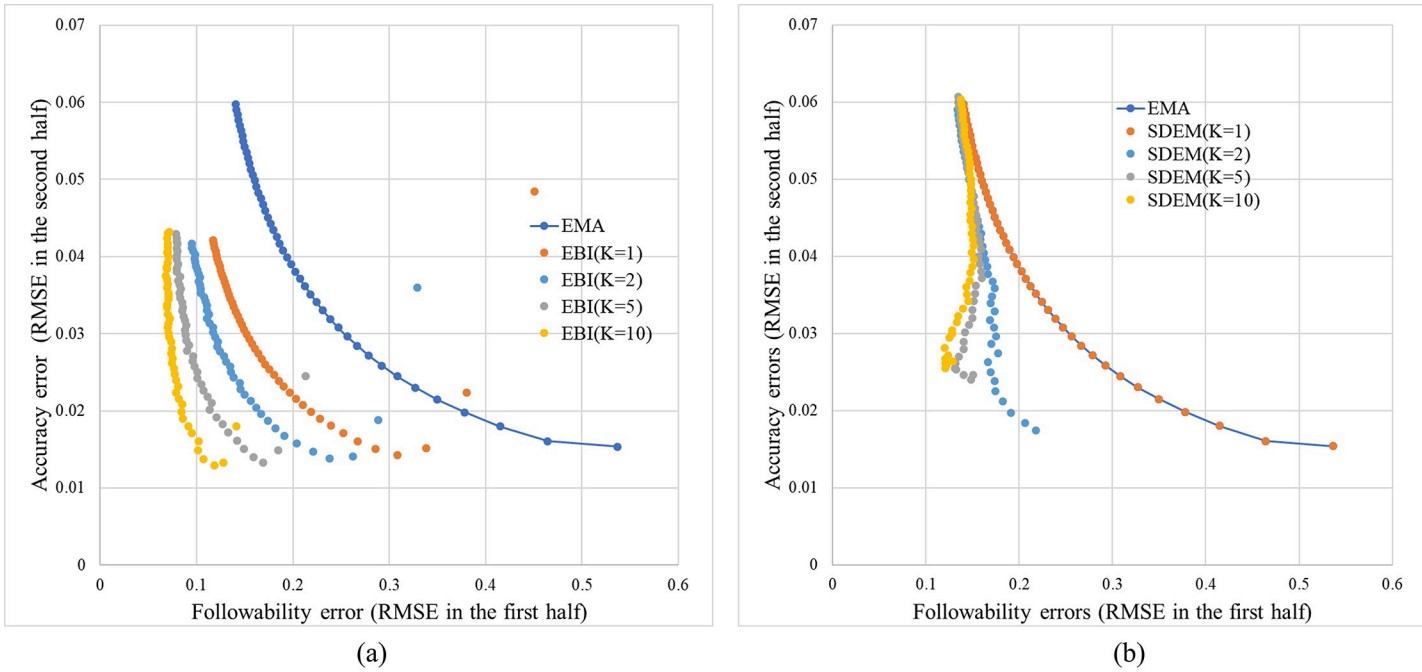

**Fig 4. Relationship between the followability errors and estimation accuracy errors.** (a) EBI. (b) SDEM.

Fig 4 shows the relationship between the followability errors and estimation accuracy errors for each method. Note that the *x*-axis and *y*-axis in this figure indicate the RMSE; therefore, the closer to the origin, the higher the estimation performance. In Fig 4(A) and 4(B), the simulations for each method were performed in the cases of $K = 1$, $K = 2$, $K = 5$, and $K = 10$. Each figure also shows the EMA results as a baseline. The simulations were performed by changing the discounting rate of each method from 0.009 to 0.15 in increments of 0.003. These values are obtained by dividing the interval from time 0 to 10000 into intervals of every 1000 steps, calculating the followability errors and accuracy errors in each interval, and averaging them.

The values shown in the figure are the average values of 100 trials with different random seeds. This also applies to Fig 5. The initial values $\mu_k^1$ of the center of each component (normal distribution) were set to random numbers generated from a uniform distribution of the range [0, 5] in both methods. Similarly, the initial values $\Sigma_k^1$ of the variance of each normal distribution were set to random numbers generated from a uniform distribution of the range [0, 0.1] in both methods.

For the EMA, it is evident that there is a trade-off, i.e., the accuracy decreases as followability increases. The EBI can modify the trade-off observed in the EMA. In the case of $K = 1$, that is, even if only the inverse Bayesian inference is used, the trade-off can be improved, but the performance is improved as the number of components is increased. SDEM can also modify the trade-off but there is no noticeable difference depending on the number of components.

Fig 5 shows the time evolution of the mean estimated value of each method where the number of components and discounting rate are selected to achieve the best performance. The correct value is also shown in the figure.

Fig 6 shows the time evolution of the estimated value of variance for each method. The figure also shows the correct value. In the EMA and SDEM, the bursts of estimates at the points where the variances change can be observed.

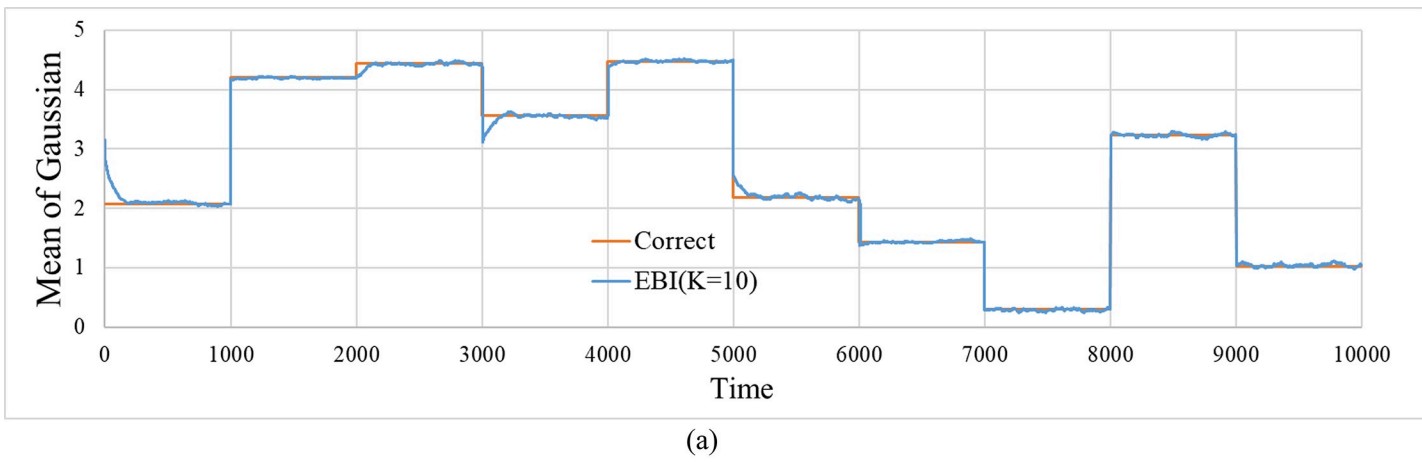

(a)

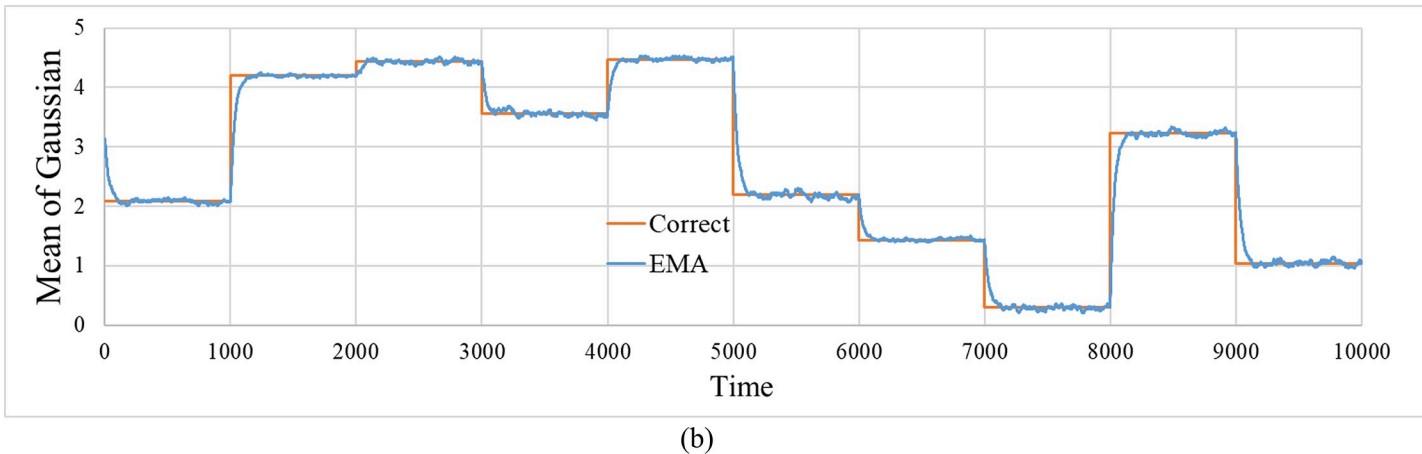

(b)

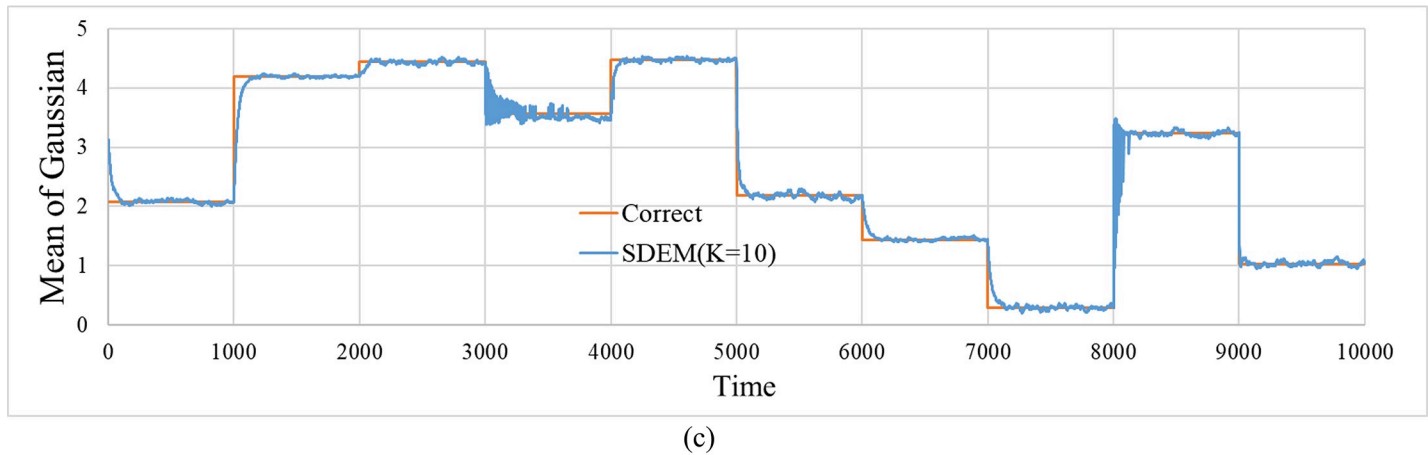

(c)

**Fig 5. Time progress of the estimated values for the mean of Gaussian.** The figure includes the correct mean. (a) Estimated values by EBI. $(\alpha, m, K) = (0.03, 0.0, 10)$. (b) Estimated values by EMA. $\beta = 0.03$. (c) Estimated values by SDEM. $(\beta, K) = (0.03, 10)$.

In the results shown above, the EBI simulations were performed for only $m = 0$. The results of EBI when $m$ is changed are shown below. The values of $\alpha$ and $m$ were shifted, with increments of 0.05 and 0.1 in the interval [0.05, 0.5] and [−2, 2], respectively, and simulations were

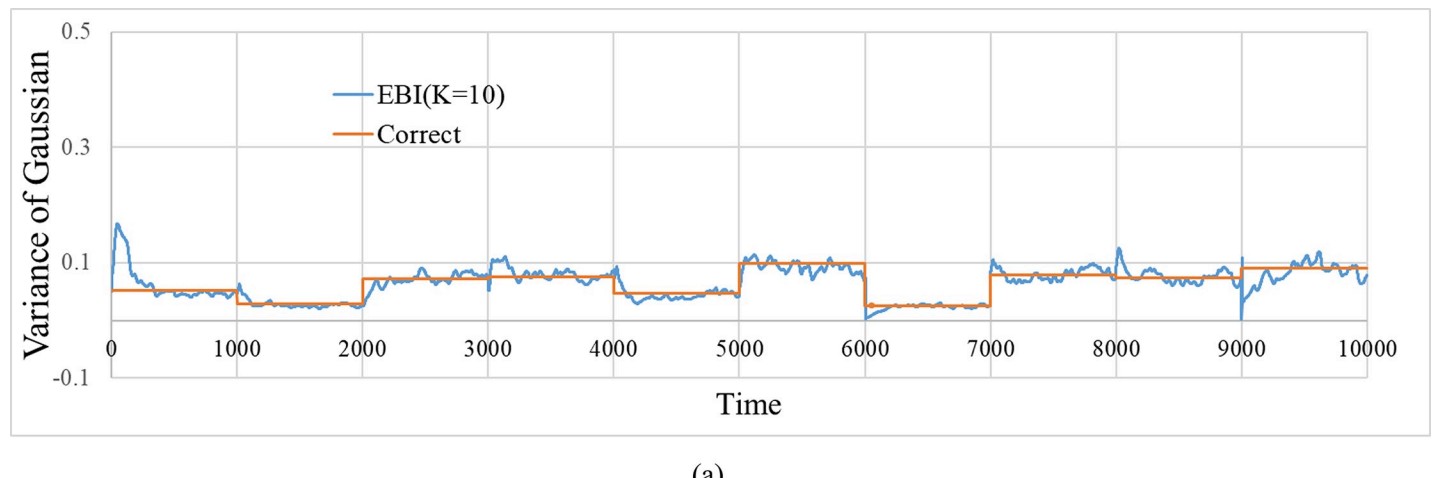

(a)

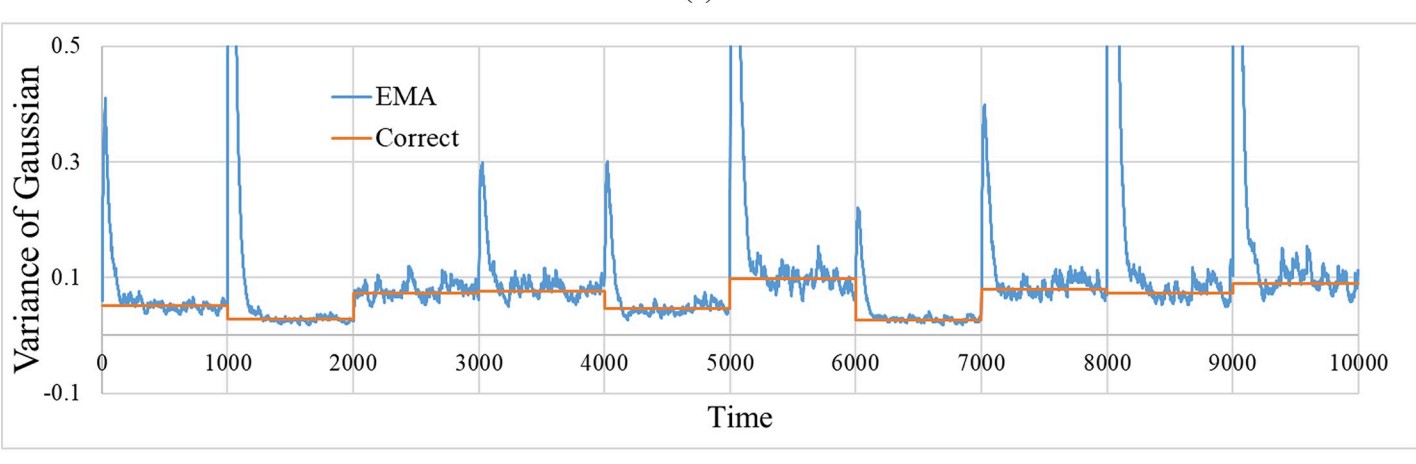

(b)

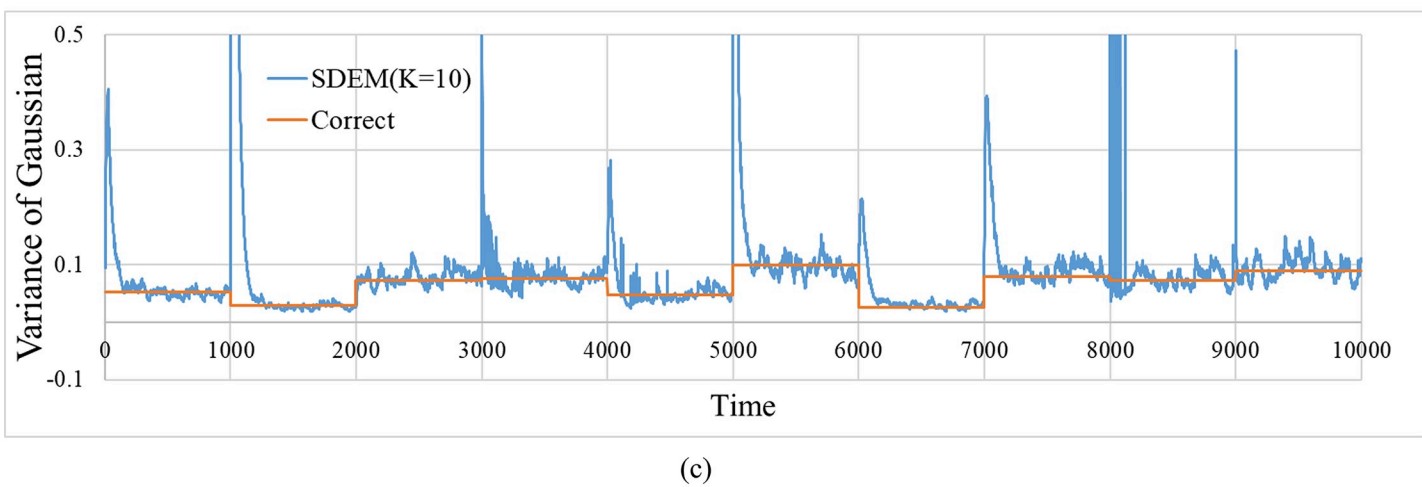

(c)

**Fig 6. Time progress for the estimated values of the variance of the Gaussian.** The figure includes the correct variance. (a) Estimated values by EBI. $(\alpha,m,K)$ = (0.03,0.0,10). (b) Estimated values by EMA. $\beta$ = 0.03. (c) Estimated values by SDEM. $(\beta,K)$ = (0.03,10).

performed to obtain the total RMSE calculated from the entire interval for each pair of parameters. Fig 7 shows the total RMSE for each pair of parameter values. In most $\alpha$ regions, the RMSE is low when $m \leq 0$. When $m$ exceeds 0, the RMSE increases.

## Discussion and conclusions

In general, if a method such as the EMA with the discounting rate is used to improve the followability to a sudden change, it is necessary to increase the discounting rate. This means that in the estimation, the recent data are weighted more extensively. That is, as long as a constant discount rate is used, a trade-off exists; the followability is improved by when the discounting rate is high, but the accuracy is reduced.

In this study, we simulated the task of estimating the distributions for data generation in a non-stationary situation wherein the distributions change suddenly. Consequently, the EBI proposed in this study successfully modified the trade-off observed in the EMA.

In addition, we compared the estimation performance of EBI with that of SDEM. The EBI showed higher estimation performance.

However, as shown in Fig 7, $m$ must be 0 or less to achieve high performance. In the literature [14], we derived $\alpha$ and $m$ that best fit the causal strength felt by humans from formula (7) and the eight types of experimental data shown in the literature [15–19]. Accordingly, the values of $\alpha$ were in the range of 0.25 to 0.6. In other words, they were far from $\alpha = 0$, which implies conditional probability. In contrast, the values of $m$ were interestingly in the range of -2.0 to -0.25, that is, were negative in all eight experiments [14]. We did not determine the cause of these negative values in this study. This is a question for further study.

As shown in Fig 6, some bursts were observed in the variance estimates by the EMA and SDEM. The changes in the mean and variance occurred simultaneously in this simulation. Therefore, the delay in following the changes in the mean may cause confusions between the changes in the mean and those in the variance.

In the EBI, various models for the hypotheses are formed by inverse Bayesian inference, even if appropriate models are not given beforehand. After some models are accumulated, rough inference is performed by switching them and fine adjustment is performed by inverse Bayesian inference, thereby achieving both followability and accuracy. The situations where both learning and inference are performed also exist in daily life. For example, in estimating the emotions of others, one cannot have a complete model for someone else's emotions because "you" are not "them." Assume that someone's facial expression suddenly changed when you estimated that the person feels happy based on your currently incomplete model. Further, assume that it was the first facial expression you saw. At this time, it is possible to think that the person's emotion has changed from joy to another emotion. However, it is also possible to consider that it is a new facial expression representing joy.

It is more difficult to detect a sign of change immediately at a time when a change is occurring than to detect a change point by looking back at the past after a change has occurred and persisted. This is because while detecting a change, the decision must be made in a situation where there is no model regarding the new stage after the change. That is, in the example above, when the next facial expression model is not completely created. Under such circumstances, while efficiently learning a model from the observed data, there is a need for a technique for making an appropriate decision using the model. In the future, such techniques can be expected to be applied for, for example, the detection of the signs of a disease.

The EBI can be regarded as introducing the effects of forgetting and learning into the Bayesian inference due to the action of exponential smoothing using $\alpha$. With the introduction of discounting rate $\alpha$, the influence of much older data is weakened. Simultaneously, $\alpha$

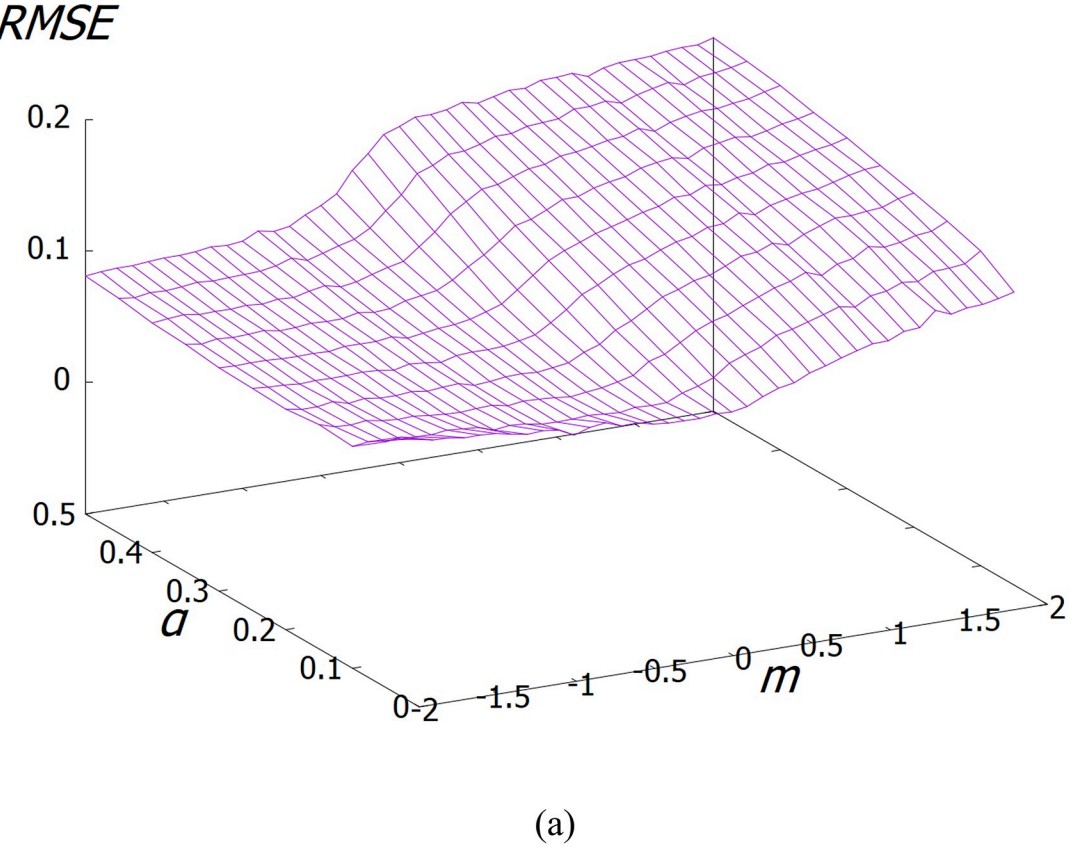

(a)

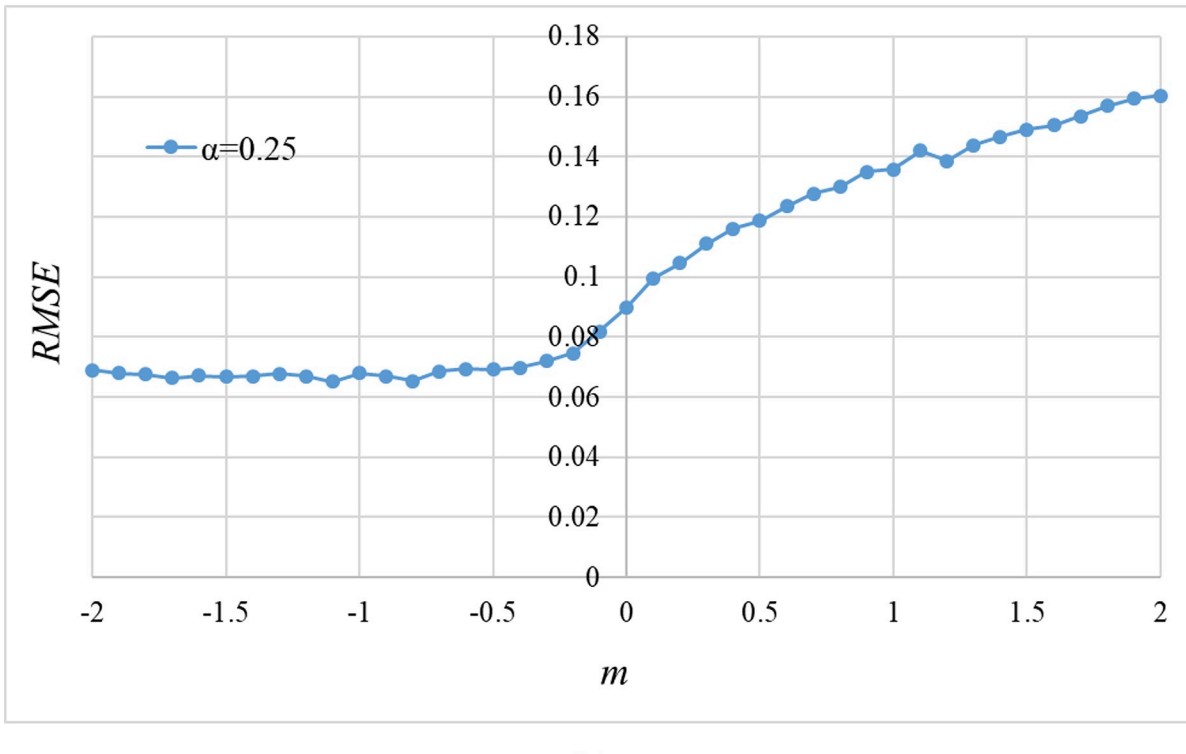

(b)

**Fig 7. RMSE for each pair of parameter values.** (a) In all cases. (b) In the case of $\alpha = 0.25$.

represents the learning rate and the learning process modifies the model for the hypothesis based on the observed data. In this framework, even with the same hypothesis, the content (model) changes over time; therefore, it is not possible to simply accumulate experiences from the past. For this reason, it is reasonable to include the effect of forgetting.

The EBI framework is very similar to the SDEM framework proposed by Yamanishi et al [7]. However, there are some differences. For example, when the history is considered for updating the weight of the Gaussian mixture distribution, there is a difference between the consideration of posterior probability or likelihood, and whether they are accumulated as addition or multiplication. In the future, we would like to clarify the difference in effectiveness due to these differences through the simulation of various tasks.

As limitations, in this simulation, only one-dimensional distribution was handled. In future work, we will extend our model to multidimensional distribution.

## Acknowledgments

We thank Editage [http://www.editage.com] for English language editing.

## Author Contributions

**Conceptualization:** Shuji Shinohara, Nobuhito Manome, Kouta Suzuki, Ung-il Chung, Tatsuji Takahashi, Hiroshi Okamoto, Yukio Pegio Gunji, Yoshihiro Nakajima, Shunji Mitsuyoshi.

**Data curation:** Shuji Shinohara.

**Formal analysis:** Shuji Shinohara.

**Funding acquisition:** Shuji Shinohara.

**Investigation:** Shuji Shinohara.

**Validation:** Shuji Shinohara.

**Visualization:** Shuji Shinohara.

**Writing – original draft:** Shuji Shinohara.

**Writing – review & editing:** Nobuhito Manome, Kouta Suzuki, Ung-il Chung, Tatsuji Takahashi, Hiroshi Okamoto, Yukio Pegio Gunji, Yoshihiro Nakajima, Shunji Mitsuyoshi.

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
