## [Decision Letter · Decision Letter 0]

26 Feb 2020

PONE-D-19-34250

A new method of Bayesian causal inference in non-stationary environments

PLOS ONE

Dear Dr. Shinohara,

Thank you for submitting your manuscript to PLOS ONE. After careful consideration, we feel that it has merit but does not fully meet PLOS ONE’s publication criteria as it currently stands. Therefore, we invite you to submit a revised version of the manuscript that addresses the points raised during the review process.

In particular, the referee found that the simulation study you presented is too limited and suggests to consider a simulation study for the normal distribution with unknown mean and variance as well and at least. In my opinion, you should also consider a study based on exponential and/or gamma distributions. The referee also suggests a series of useful minor revisions that I hope should further improve your manuscript. Please, see the attached .pdf file.

We would appreciate receiving your revised manuscript by Apr 11 2020 11:59PM. To enhance the reproducibility of your results, we recommend that if applicable you deposit your laboratory protocols in protocols.io, where a protocol can be assigned its own identifier (DOI) such that it can be cited independently in the future. For instructions see: http://journals.plos.org/plosone/s/submission-guidelines#loc-laboratory-protocols

We look forward to receiving your revised manuscript.

Kind regards,

Enrico Scalas, Ph.D.

Academic Editor

PLOS ONE

Journal Requirements:

The authors have declared that no competing interests exist.

We note that one or more of the authors are employed by a commercial company: SoftBank Robotics Group Corp.

Reviewers' comments:

Reviewer's Responses to Questions

**Comments to the Author**

1. Is the manuscript technically sound, and do the data support the conclusions?

Reviewer #1: Partly

2. Has the statistical analysis been performed appropriately and rigorously? 

Reviewer #1: Yes

3. Have the authors made all data underlying the findings in their manuscript fully available?

Reviewer #1: Yes

4. Is the manuscript presented in an intelligible fashion and written in standard English?

Reviewer #1: Yes

5. Review Comments to the Author

Reviewer #1: There is a major issue concerning the simulation study, and some minor comments to be addressed before publication. See the attached report for details.......................................................................

6. PLOS authors have the option to publish the peer review history of their article (what does this mean?). If published, this will include your full peer review and any attached files.

Reviewer #1: No

---

## [Author Response · Author response to Decision Letter 0]

27 Mar 2020

Thank you for your careful peer review and helpful comments.

Major comments.

I have basically only one major comment, concerning the simulation study. The authors present a simulation study for the expected value of a Normal distribution with known variance. This is a very strong limitation, since usually the normal distribution of an underlying process, as in the proposed application, has the two parameters both unknown. The authors claim at the end of the paper that the estimation of the variance and the extension to multi-dimensional parameters will be the subject of a subsequent paper, but I would suggest to add to the present paper some more scenarios in the simulation study, such as the mean and variance of a Normal distribution. Also the estimation of the expected value of other distributions, such as the exponential distribution should be considered.

We have made the recommended changes to ensure that both the mean and variance can be estimated simultaneously. We have divided the section “Testing a normal distribution model” into two subsections, i.e., “Mean value estimation using normal distribution” and “Variance estimation using gamma distribution”; the latter describes the variance estimation. For the variance estimation, we have introduced a gamma distribution and found the solution using Lambert's W function. The results of the variance estimation are shown in Fig.6. With respect to this change, the whole manuscript has been modified to be consistent.

Minor remarks.

 Some sentences in the text must be rewritten. For instance, the first sentence of the introduction: “Bayesian inference deduces…” should be “The aim of Bayesian inference is to deduce …”. Also in line 156, “becomes a tautology” is not logically correct.

Thank you for pointing this out.

We have added “The aim” and “to” in line 38.

“Bayesian inference deduces…”



“The aim of Bayesian inference is to deduce …”

We have modified line 170.

“…, formula (12) becomes a tautology as shown below:”



“…, does not change by formula (14), as shown below.”

 Although numerical aspects are relevant in this paper, the use of 0.0 for 0 and 1.0 for 1 should be avoided.

We modified 0.0, 1.0, and 5.0 in the text to 0, 1, and 5, respectively.

 Lines 135-140. The meaning of the parameter m must be discussed.

Thank you for pointing out this important point.

We have interpreted your comment in two ways. One is that we thought you meant our explanation of parameter m was too brief. Therefore, we added lines 145-153 to clarify the meaning of parameter m. The other is that we thought you meant our treatment of the case where m = 0 was too brief, as you referred to lines 135-140. Therefore, we added lines 134-139 that contain the proof for the case m = 0. 

 Formula (10) in line 141 must be proved in full details. I do not see that it is an obvious consequence of the preceding formulas.

We have added the details for equation conversion in formula (12).

 Line 205. The use of ε = 10-300 must be checked with care. In several frameworks, such a value is too small for preventing from zero probabilities.

Thank you for pointing out this important point. 

We have changed ε = 10-300 to ε = 10-10 in line 221. 

 Figure 1 can be removed.

According to this suggestion, we have removed Fig. 1.

 The section “EM algorithm for Gaussian mixture learning” contains known material and can be considerably shortened.

Thank you for pointing this out.

We have deleted the section “EM algorithm for Gaussian mixture learning.” Moreover, we have included some necessary items in the section “Sequential Discounting Expectation-Maximisation Algorithm (SDEM).”

---

## [Decision Letter · Decision Letter 1]

8 May 2020

A new method of Bayesian causal inference in non-stationary environments

PONE-D-19-34250R1

Dear Dr. Shinohara,

We are pleased to inform you that your manuscript has been judged scientifically suitable for publication and will be formally accepted for publication once it complies with all outstanding technical requirements.

With kind regards,

Enrico Scalas, Ph.D.

Academic Editor

PLOS ONE

Additional Editor Comments (optional):

The referee is satisfied with your revisions.

Reviewers' comments:

Reviewer's Responses to Questions

**Comments to the Author**

1. If the authors have adequately addressed your comments raised in a previous round of review and you feel that this manuscript is now acceptable for publication, you may indicate that here to bypass the “Comments to the Author” section, enter your conflict of interest statement in the “Confidential to Editor” section, and submit your "Accept" recommendation.

Reviewer #1: All comments have been addressed

2. Is the manuscript technically sound, and do the data support the conclusions?

Reviewer #1: Yes

3. Has the statistical analysis been performed appropriately and rigorously? 

Reviewer #1: Yes

4. Have the authors made all data underlying the findings in their manuscript fully available?

Reviewer #1: Yes

5. Is the manuscript presented in an intelligible fashion and written in standard English?

Reviewer #1: Yes

6. Review Comments to the Author

Reviewer #1: The authors have addressed all my concerns on the first version. I am satisfied with this version and I recommend acceptance.

7. PLOS authors have the option to publish the peer review history of their article (what does this mean?). If published, this will include your full peer review and any attached files.

Reviewer #1: No

---

## [Editor Report · Acceptance letter]

12 May 2020

PONE-D-19-34250R1 

A new method of Bayesian causal inference in non-stationary environments 

Dear Dr. Shinohara:

I am pleased to inform you that your manuscript has been deemed suitable for publication in PLOS ONE. Congratulations! Your manuscript is now with our production department. 

With kind regards,

on behalf of

Professor Enrico Scalas 

Academic Editor

PLOS ONE